# Ozone Concentration Levels in Urban Environments—Upper Silesia Region Case Study

**DOI:** 10.3390/ijerph18041473

**Published:** 2021-02-04

**Authors:** Joanna Kobza, Mariusz Geremek, Lechosław Dul

**Affiliations:** 1Department of Public Health, School of Health Sciences, Medical University of Silesia, Piekarska 18, 41-902 Bytom, Poland; mgeremek@sum.edu.pl; 2Department of Epidemiology and Biostatistics, School of Health Sciences, Medical University of Silesia, Piekarska 18, 41-902 Bytom, Poland; ldul@sum.edu.pl

**Keywords:** population health, ozone, environmental policy, air pollution

## Abstract

Although ozone (O_3_) plays a crucial role in screening the Earth’s surface and lower atmosphere layers from the ultraviolet radiation, troposphere ozone is proven to have negative health effects on the human body and is one of the greenhouse gases. The objective of this study was to perform a measurement-based assessment for determining whether the concentration of ozone is within admissible limits, or exceeded, in Silesia Province and does not pose a threat to the local population. The data provided by the Voivodship Inspectorate for Environmental Protection in Katowice were used in the analysis. The received data constitute the result of 8-h measurements of concentrations of ozone at selected air monitoring stations of the Silesian province. The locations of three monitoring stations were found to be useful for the aim of this research; one site is situated in a rural background area; another one is located in a medium-sized city and the Katowice station is representative for an urban background situation. We used cluster analysis, weighted pair group method using arithmetic averages (WPGMA) and Chebyshev distances to test the hypothesis and compare empirical distributions in the general population. The alarm level has not been exceeded in indicated measurements stations in Silesian Voivodship in the period 2015–2017 (averaging time 1 h: 240 µg/m^3^ for 3 h). The target level was exceeded in 2015 at all three measurements stations and in the following years at one station (in Zloty Potok, 2016, and in Katowice, 2017). Each year, the largest number of exceedances occurred in August. The results clearly indicate a lack of hazards for the general population’s health in terms of increased concentrations of ozone in the city centers and outside. The results confirm that environmental conditions (i.e., landform, the area surrounding monitoring station) have a significant influence on the ozone level.

## 1. Introduction

Air pollution is a leading environmental factor that influences human life; therefore, negative health consequences of the low quality of air are cited as main public health concerns [1,2]. Long-term exposure to a higher ozone concentration is related to a decrease in life expectancy [3]. Recent research highlighted that in 2013, exposure to ozone resulted in 16,000 premature deaths and 192,000 years of life lost (YLL) [4]. Stricter national and international air quality guidance could reduce ozone-related premature mortality [5]. Although ozone (O_3_) plays a crucial role in screening the Earth’s surface and lower atmosphere layers from ultraviolet radiation, troposphere ozone is proven to have adverse health effects on the human body and is one of the greenhouse gases [1,2]. Ozone is an allotropic form of oxygen; it consists of three oxygen atoms (O_3_). The third atom of oxygen makes ozone a firm photochemical oxidant. Ozone in the upper stratum of the atmosphere—the stratosphere—arises as a result of the interaction of solar radiation with oxygen molecules (O_2_). Due to the interaction of the sun, these molecules break down into two oxygen atoms (O). Then, the oxygen atoms react with molecular oxygen and form ozone molecules (O_3_) [6]. This ozone is called stratospheric ozone or the ozone layer. Ozone in the stratosphere almost completely absorbs ultraviolet radiation with a wavelength below 295 nm, which is harmful to living organisms [5]. Due to the ozone layer, only a small percentage of biologically active UV radiation reaches the Earth’s surface [7]. Tropospheric ozone is an effect of the chemical reaction of volatile organic compounds (VOCs) such as nitrogen oxide (NO_x_) [2]. There is an important role of the process of photochemical oxidation of hydrocarbons, carbon monoxide and other ozone precursors in the presence of nitrogen oxide in the ozone formation process [8]. Photochemical reaction of nitrogen oxides and volatile organic compounds is accelerated by the high temperature of air and sunshine. The highest concentrations of ozone at the surface of the Earth occur in spring and summer, outside the city or suburbs. The release of ozone precursors into the atmosphere is mostly the effect of industrialization and other anthropogenic sources (i.e., vehicle traffic and combustion processes) [9].

The levels of tropospheric ozone could also depend on meteorological or climatic changes [10]. At both the national and regional scales, meteorological conditions are the dominant drivers of ozone production [11]. Long-range transport of tropospheric ozone and its precursors can occur [12,13,14]. Its concentrations in Poland largely depend on ozone concentrations in the air, mainly from southern and southwestern Europe [6]. The highest ozone levels in Poland are reported in the Upper Silesia agglomeration and the Sudetes and Tatra Mountains in the southern part of the country. On hot, sunny days, the highest concentrations of ozone occur during the afternoons [7]. Ozone is one of three key pollutants taken into account in when calculating the Common Air Quality Index (CAQI), which is used for comparison and classification of the air quality in European Union countries.

## 2. Aim

The aim of the study was to assess whether the ozone concentration level was within admissible limits or was in excess in the industrial agglomeration of Silesian Voivodship (with a population of about five million inhabitants) and, therefore, poses a threat to the local population together with the levels of PM_2.5_ and PM_10_, significantly exceeded at the same time. Data have been compiled to describe trends of ozone concentration level for the period of 2015–2017.

## 3. Methodology

The ozone concentration levels were collected within the project of State Environmental Monitoring and provided by the Voivodship Inspectorate for Environmental Protection in Katowice. Based on one-hour concentrations, mean values of three fields—daily average, the highest 8-h moving average (normalized value) and the highest one-hour value of ozone concentrations during the day—were calculated for each day (for 24-, 48- and 72-h periods).

The 8-h ozone concentration was measured at 3 air monitoring stations located in different parts of the Silesian Voivodship (Katowice, Bielsko-Biała and Złoty Potok). The air monitoring stations collected data, with referential methods specified in the domestic law, on the assessment of levels of substances in the air [15]. The descriptions of measurement methods applied for the registration of ozone are presented below in the part on the measurement stations. In Poland, there are approximately 100 ground-level ozone measurement stations located throughout the country. Ground-level ozone measurements are based on ultraviolet photometry using automatic analyzers. The air intakes for sampling the air are usually located at the height of 2.5 to 4 m above the ground. Ozone analyzers are located in special air-conditioned containers with a constant temperature of approximately 20–23 °C. The ultraviolet photometry method is the reference method for measuring ozone in the air. The locations of the 3 monitoring stations were found to be useful for this research; one site is situated in a rural background area; another one is located in a medium-sized city and the Katowice station is representative of the urban background situation. We used cluster analysis, weighted pair group method using arithmetic averages (WPGMA) and Chebyshev distances to verify the relationship between the area surrounding the location of the monitoring stations and the number of days on which ozone concentrations exceeded the regulatory standards. We also identified the main sources of ozone and the levels of ozone precursors emissions in the period 2015–2018 in the Upper Silesia urban area.

### 3.1. Measuring Stations

The measurements were carried out in the measuring stations, which are part of the net of the measuring stations of the State Environmental Protection program in Poland. The measurements were carried out from 1 January 2015 to 31 July 2017 (Figure 1).

The first station was located in Katowice, 40-844, Kossutha Street 6. Measurement type: automatic and manual. Name: SL09KA. International code: PL0008A. Type: Urban background station. Measurement zone: Silesian agglomeration. Measurement target: human health protection.

The second station was located in Złoty Potok, near Częstochowa. Measurement type: automatic and manual. Name: SL03ZP. International code: PL0243A Urban background station. Measurement zone: Silesian agglomeration. Measurement target: plant protection.

The third station was located in Bielsko-Biała, 43-300, Kossak-Szczuckiej Street 19. Measurement type: automatic and manual. Name: SL15BB. International code: PL0234A Urban background station. Measurement zone: Silesian agglomeration. Measurement target: human health protection.

### 3.2. Health Risk

Exposure to ozone is a significant health issue in both developed and developing countries. In 2015, ozone exposure caused 254,000 premature deaths globally and 16,400 in the EU [16,17]. Even short-term exposure to ozone within values of air quality standards could lead to increased risk of mortality among older adults [18]. Ozone is a significant oxidant, which could interact with cells of different human systems (i.e., the respiratory system and lungs, the cardiovascular system and eye tissues) [11].

Higher concentrations of ozone in the air could lead to presbyopia among old adults, the inflammation of eye tissues and could also exacerbate the negative effects of allergic reactions on the ocular surface [19,20].

Ozone exposure is also a risk factor for respiratory diseases (including the worsening of asthma symptoms and reduced lung function). In 2015, 9–23 million emergency room visits due to asthma (which is the most common chronic respiratory disease in the world with about 358 million people affected) were attributable to ozone [4,21]. Long-lasting exposure to ozone could also increase asthma hospital admissions among children [22,23]. Young children and those coming from lower socioeconomic groups had a higher risk of asthma than other children at the same level of ozone [24]. Exposure to ozone increases the burden of chronic obstructive pulmonary disease (COPD) [25]. It is estimated that in 2015 exposure to ozone resulted in 4.1 million DALYs from COPD [17]. Exposure to ozone could also influence the incidence of different respiratory system symptoms such as coughing, throat irritation and chest tightness [26].

As an effect of exposure to ozone, the human body defends itself against high concentrations of ozone by reducing the amount of inhaled oxygen, which could affect the severity of cardiovascular diseases [5]. According to the American Cancer Society cohort study, summer ozone levels were associated with both total and cardiopulmonary mortality [27]. Oxidative stress, increased level of inflammatory cytokines and endothelial dysfunction could influence the risk of cardiovascular diseases as an effect of ozone exposure [28]. Short-term exposure to ozone was connected with a higher risk of out-of-hospital cardiac arrest, cardiovascular and respiratory mortality and a higher possibility of death in patients previously hospitalized due to myocardial infarction and those with their first manifestation of this disease [29,30]. Ozone exposure was also associated with increased number of years of life lost (YLL) from hypertension. The association was higher in elderly individuals born in the months of higher ozone concentration [31]. Although the increase in cancer risk differs due to weight, age, sex and health-related behaviors, long-term exposure to ozone increases cancer risk (especially lung cancer) [32,33]. Ozone can also cause somnolence, headaches and tiredness as well as decreases in blood pressure [6]. Recent studies suggest that there is a connection between exposure to ozone and preterm birth and low birth weight of newborns [34,35]. Ozone could also cause the onset of infantile eczema [36]. Maternal exposure to ozone was associated with the development of hypertensive disorders in pregnancy [37].

### 3.3. WHO Recommendations

According to WHO guidelines, the daily concentration of ozone should not exceed the level of 100 µg/m^3^ (daily maximum 8-h mean) (Table 1). Time-series studies show a 0.3–0.5% rise in daily mortality for every 10 µg/m^3^ increase in 8-h ozone concentrations above an approximate level of 70 µg/m^3^ [38]. It is worth noting that the European Union ozone concentration standards are less restrictive than the recommendations contained in the WHO guidelines.

### 3.4. National Policy

According to the regulation by the Minister of Environment on 24 August 2012 on the levels of certain substances in the air [13], the following definitions of the levels of concentrations for ozone were adopted in this paper.

Target level—The level of substance in the air which is to be achieved within a certain time using economically justified technical and technological measures; it is determined to avoid, prevent or limit the harmful effects of the substance on the health of the population or the environment. The target level for ozone is 120 µg/m^3^ (maximum daily 8-h mean), the target value to be met as of 01.01.2010. Permitted exceedance each year is 25 days.

Information level—The concentration level of certain substances in the air proven to be a threat to the health of the people whose health could be at risk after short-term exposure; thus, adequate information is required. Information level for ozone is 180 µg/m^3^ (1-h average). 

Alarm level—The concentration level of certain substances in the air proven to be a threat to the health of the whole community after short-term exposure; thus, immediate action must be undertaken by EU member states. Alarm level for ozone is 240 µg/m^3^ (1-h average). 

Occurrence in the air of ozone at a concentration exceeding the alarm level negatively affects the entire population. In this case, people from vulnerable groups, i.e., sick people, especially those with respiratory diseases, children and the elderly, should avoid being in the open, and other persons should limit being outside to the minimum necessary.

### 3.5. European Standards of the Ozone Concentration

The ozone concentration level accepted in the European Union is 120 µg/m^3^ (maximum daily 8-h mean), with permitted exceedances on 25 days averaged over 3 years (target value to be met as of 1.1.2010) [39]. The European ozone regulation No. 1005/2009 on substances that deplete the ozone layer constitutes the legal basis for the protection of the ozone layer within the European Union [40]. The rule has two purposes: to execute the obligations of the Montreal Protocol (1987) on substances that deplete the ozone layer, which all EU member states have made a formal commitment to respect, and to ensure a higher level of protection in some areas in the European Union than adopted in the protocol [41]. In 2017, the European Commission started an evaluation of the Ozone Regulation to investigate its implementation and execution in member states, which is due to be completed in 2019. The Ozone Regulation is the basis for other EU regulations concerning limitation of substances that deplete the ozone layer, i.e., Commission Regulation No. 537/2011 [42], Commission Regulation No. 291/2011 [43] and Commission Decision No. 2010/372/EU [44]. The European Commission is responsible for not only monitoring but also reporting the data to the United Nations Environment Program (UNEP) Ozone Secretariat on the utilization of ozone-depleting substances in all EU member states according to the Montreal Protocol recommendations. The rules concerning the number and location of sampling points for the assessment of ambient air quality, data validation and reference methods for the measurement of ozone are regulated by Directive 2015/1480/EC of 28 August 2015 [45]. The questions of the reciprocal exchange of information and reporting on ambient air quality are regulated by the Commission Implementing Decision of 12 December 2011, which put into effect Directives 2004/107/EC and 2008/50/EC of the European Parliament and of the Council (No C(2011) 9068) [46] (Table 2).

## 4. Results

### 4.1. Measurement Station in Bielsko-Biała SL15BB

Statistical analyses indicated exceedances of the target level of ozone (maximum daily 8-h mean equal or more than 120.4 μg/m^3^) 32 times (days) registered at the measurement station in 2015.

Statistical analyses indicated exceedances of the target level of ozone (maximum daily 8-h mean equal or more than 120.4 μg/m^3^) eight times (days) registered at the measurement station in 2016. 

Statistical analyses indicated exceedances of the target level of ozone (maximum daily 8-h mean equal or more than 120.4 μg/m^3^) 15 times (days) registered at the measurement station in 2017 (Figure 2, Figure 3 and Figure 4). 

### 4.2. Measurement Station in Katowice SL09KA

Statistical analyses indicated exceedances of the target level of ozone (maximum daily 8-h mean equal or more than 120.4 μg/m^3^) 32 times (days) registered at the measurement station in 2015.

Statistical analyses indicated exceedances of the target level of ozone (maximum daily 8-h mean equal or more than 120.4 μg/m^3^) 15 times (days) registered at the measurement station in 2016; according to the law, we did not observe exceedances of the target level of ozone in 2016.

Statistical analyses indicated exceedances of the target level of ozone (maximum daily 8-h mean equal or more than 120.4 μg/m^3^) 32 times (days) registered at the measurement station in 2017 (Figure 5, Figure 6 and Figure 7). 

### 4.3. Measurement Station in Zloty Potok SL03ZP

Statistical analyses indicated exceedances of the target level of ozone (maximum daily 8-h mean equal or more than 120.4 μg/m^3^) 61 times (days) registered at the measurement station in 2015.

Statistical analyses indicated exceedances of the target level of ozone (maximum daily 8-h mean equal or more than 120.4 μg/m^3^) 30 times (days) registered at the measurement station in 2016. 

Statistical analyses indicated exceedances of the target level of ozone (maximum daily 8-h mean equal or more than 120.4 μg/m^3^) 25 times (days) registered at the measurement station in 2017. According to the law, we did not observe exceedances of the target level of ozone in 2017.

The alarm level was not exceeded at the indicated measurement stations in Silesian Voivodship in the period 2015–2017 (averaging time 1 h: 240 μg/m^3^ for 3 h). The target level was exceeded in 2015 at all three measurement stations and in the following years at one station (in Zloty Potok in 2016 and in Katowice in 2017). Each year, the largest number of exceedances occurred in August (Figure 8, Figure 9 and Figure 10). 

The results indicate a relationship between the area surrounding and the location of monitoring stations and the number of days on which ozone concentrations exceeded the regulatory standards. The year of measurement also affects the number of days exceeding the target ozone level.

In 2015, the highest number of days exceeding the O_3_ target level was observed for all air quality monitoring stations, with a certain exception being the SL09KA measuring station. In 2016, for all monitoring stations, there was a clear decrease in the number of days exceeding the O_3_ target level. The year 2017 shows an approx. two-fold increase in the number of days exceeding the O_3_ target level compared to 2016 for the SL15BB and SL09KA measuring stations. For the SL03ZP measuring station, in 2017, the number of days exceeding the O_3_ target level decreased from 30 days to 25 days (Table 3).

We used cluster analysis, weighted pair group method using arithmetic averages (WPGMA) and Chebyshev distances to test this hypothesis and compare empirical distributions in general population (Table 4 and Table 5).

The obtained results confirmed that environmental conditions (i.e., landform, the area surrounding monitoring station) have a significant influence on the ozone level. The results also showed that the ozone concentration level was less dependent from PM_10_ or PM_2.5_. The PM_10_ and PM_2.5_ levels in the area surrounding the monitoring station in Złoty Potok were significantly lower than in the area surrounding the Katowice measuring station. Nevertheless, the number of days on which ozone concentrations exceeded the regulatory standards at the Złoty Potok measuring station was much higher than in the Katowice measuring station.

In the Upper Silesia urban area, the main sources of ozone are industrial point sources (i.e., factories and coal-planted power plants), burning coal for heating of households (residential plants), transportation (especially road vehicles) and agricultural activities (emissions from manure management, application of fertilizers, field burning of agricultural residues). Forest areas, especially dominated by conifer forests, are also natural sources of biogenic emissions of volatile organic compounds (i.e., terpenes and isoprene). We compared the levels of ozone-forming gas (ozone precursors) emissions in the period 2015–2018 (Table 6, Table 7, Table 8, Table 9 and Table 10). An increase in NO_x_ and CO emissions between 2015 and 2018 was observed as well as a decrease in SO_2_, NH_3_ and non-NH_3_ volatile organic compound emissions at the same time [47,48].

The classification used to assess emission of ground level ozone precursors in the Silesian voivodship in 2018 was defined within the framework of the Selected Nomenclature for sources of Air Pollution (SNAP) 97 [49].

## 5. Discussion

Ozone is formed near the surface of the Earth, in particular in periods of high temperatures and in high insolation in the presence of other substances such as PM, NO and CO, which may have synergistic effects and negatively affect the health of the population [50,51]. The current climate change context is also crucial [52,53,54]. In recent years, we have observed a rise in the average air temperature, which may cause concern regarding increased exposure to ozone emissions [55]. At all three monitoring stations, the hottest month in the year was July, and the average temperature during this month differs; for Katowice, it is 19 degrees; for Zloty Potok, it is 16.7 degrees; and for Bielsko-Biała, it is 18.6 degrees. Here, 2015 was the warmest year in the period under review, while the average annual temperature exceeded 10 degrees, which could partly explain why we noted that year as having the most exceedances in all three measurement stations [56].

Furthermore, the growth in the number of vehicles, especially in bigger cities, and the corresponding emissions create challenges for improving air quality. According to the report of UN-Habitat, cities produce more than 60% of greenhouse gas emissions and consume 78% of the world’s energy; nevertheless, they are responsible for less than 2% of the Earth’s surface [57]. Some research indicated that people are more exposed to high levels of O_3_ in rural areas than citizens living in cities [58]. Other research also confirmed new factors such as sex, obesity and socioeconomic status, which may also increase the risk of O_3_-related health effects [59,60]. 

Although the alarm level was not exceeded at all three monitoring stations in the period of 2015–2017, the target level (120 µg/m^3^) was exceeded many times every year during the summer. The improvement of air quality could provide significant health benefits. Experts estimate that limitations of O_3_ levels to 98 µg/m^3^ would prevent approximately 400 annual deaths, and the greatest health impact was observed in the age group over 65 and in mortality due to cardiovascular and cardiopulmonary diseases [61]. 

It is also important to note that over the past 20 years, the threat to the population decreased in many European countries, including France in the period between 1999 and 2012, demonstrating the effectiveness of European control strategies and regulations [57]. The air in Poland is significantly more polluted than that in most of the European Union countries [62]; therefore, changes in meteorological conditions, especially to temperature and humidity anticipated in the nearest future, should be taken into account. That is why it is necessary to define more restrictive standards concerning ozone for human health and environmental protection in Poland.

Further and more extensive research in the region is also recommended, using, for example, machine learning techniques, as they can be useful in achieving high prediction accuracy of ground-level ozone (i.e., to predict the period of the year with the highest ozone concentration in a defined geographical area) [63,64,65].

## 6. Conclusions

Providing evidence-based information which could serve policy-making and be accessible by the media and the general public seems to be essential from the public health point of view. The results of this study provide such evidence in the territory of the Silesian Voivodship (population of five million people). The results clearly indicate no hazards for the health of the general population in terms of increased concentrations of ozone in the city centers and the surrounding areas. The results confirm that environmental conditions (i.e., landform, the area surrounding monitoring station) have a significant influence on the ozone level. Concerning ozone-forming gases’ emission, an increase in NO_x_ and CO emissions in the period 2015–2018 and, at the same time, a decrease in SO₂, NH_3_ and non-NH_3_ volatile organic compound emissions were observed. We identified the main sources of ozone in the Upper Silesia urban area, which are residential plants and point sources. NO_x_ and CO control would reduce the risk of high ozone concentration in the summer, especially in the central and southern parts of the country (including Upper Silesia).

## Figures and Tables

**Figure 1 ijerph-18-01473-f001:**
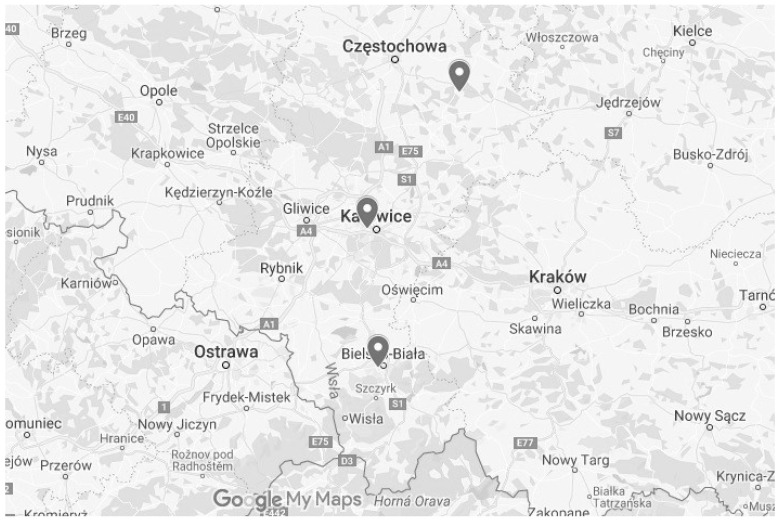
Map. The locations of measurement sites in Bielsko-Biała, Katowice and Zloty Potok in the Upper Silesian agglomeration used in the study.

**Figure 2 ijerph-18-01473-f002:**
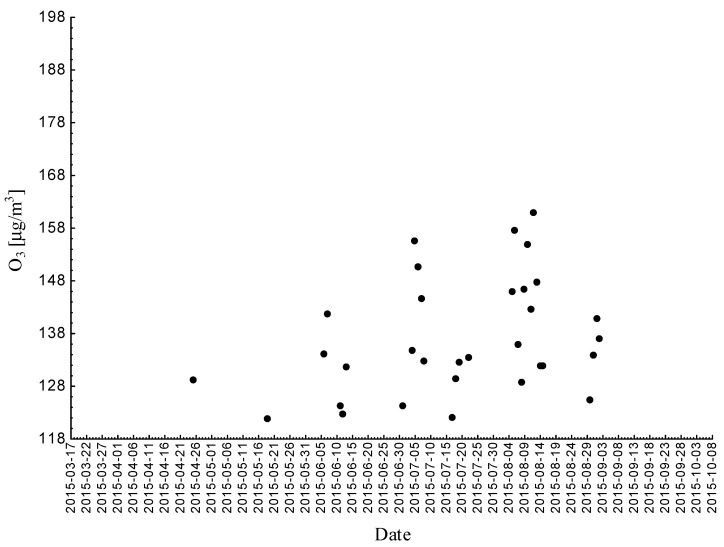
Maximum daily 8-h mean ozone concentration not less (equal to or more than) than 120.4 μg/m^3^ registered at the measurement station in Bielsko-Biała (SL15BB) in 2015.

**Figure 3 ijerph-18-01473-f003:**
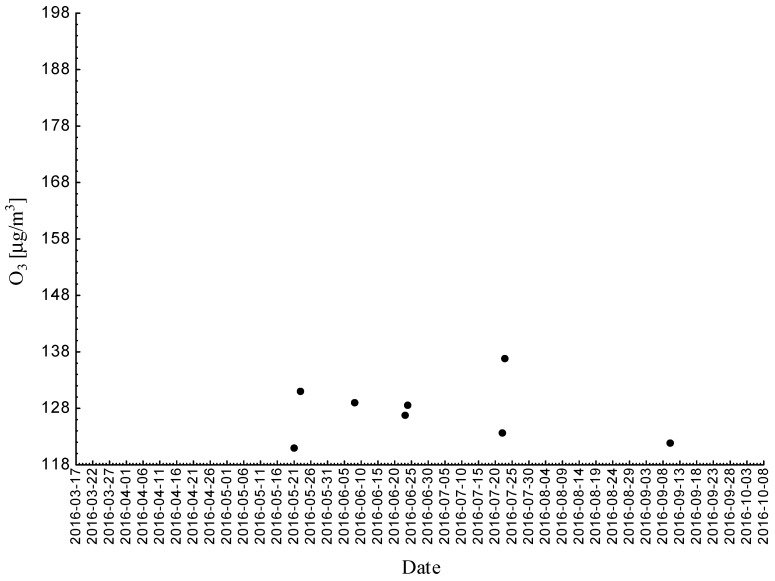
Maximum daily 8-h mean ozone concentration not less (equal to or more than) than 120.4 μg/m^3^ registered at the measurement station in Bielsko-Biała (SL15BB) in 2016.

**Figure 4 ijerph-18-01473-f004:**
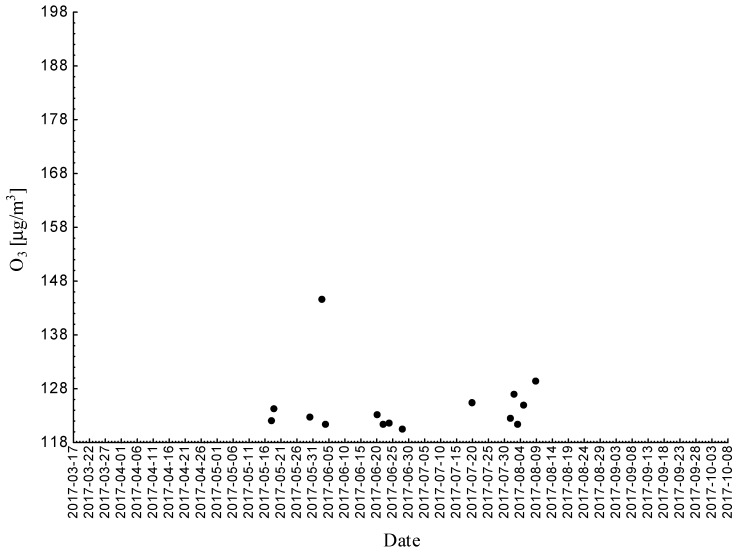
Maximum daily 8-h mean ozone concentration not less (equal to or more than) than 120.4 μg/m^3^ registered at the measurement station in Bielsko-Biała (SL15BB) in 2017.

**Figure 5 ijerph-18-01473-f005:**
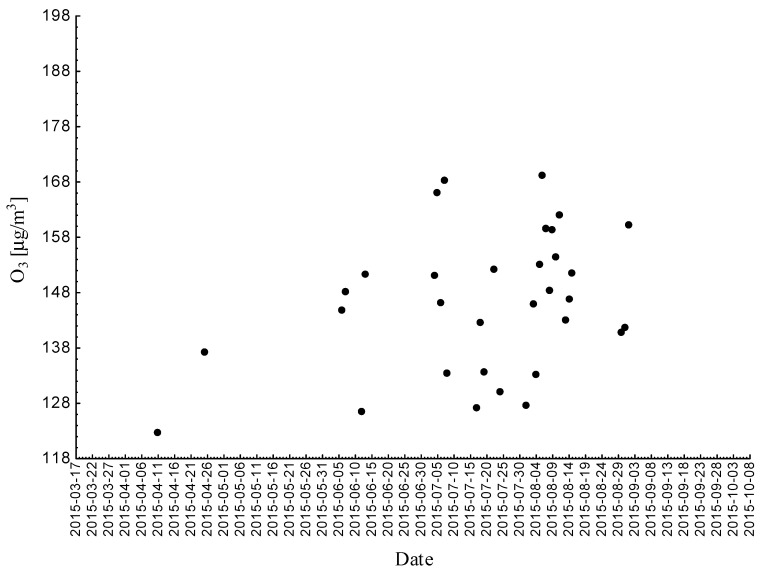
Maximum daily 8-h mean ozone concentration not less (equal to or more than) than 120.4 μg/m^3^ registered at the measurement station in Katowice (SL09KA) in 2015.

**Figure 6 ijerph-18-01473-f006:**
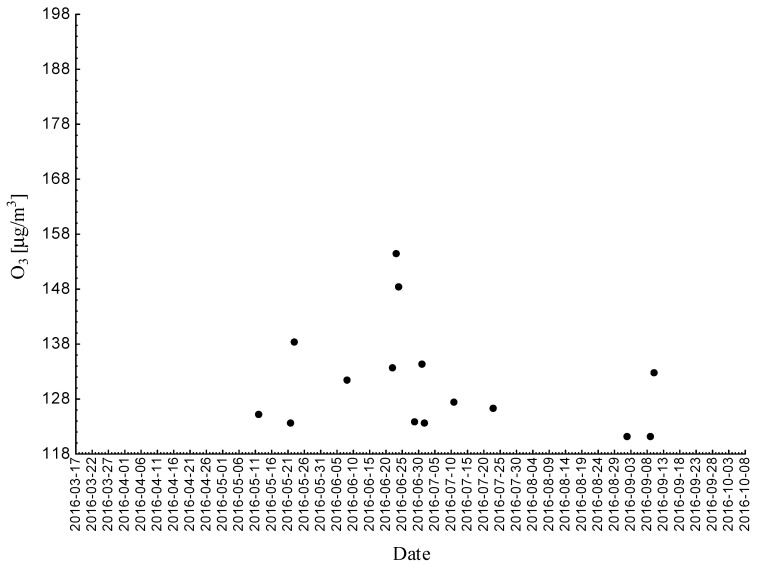
Maximum daily 8-h mean ozone concentration not less (equal to or more than) than 120.4 μg/m^3^ registered at the measurement station in Katowice (SL09KA) in 2016.

**Figure 7 ijerph-18-01473-f007:**
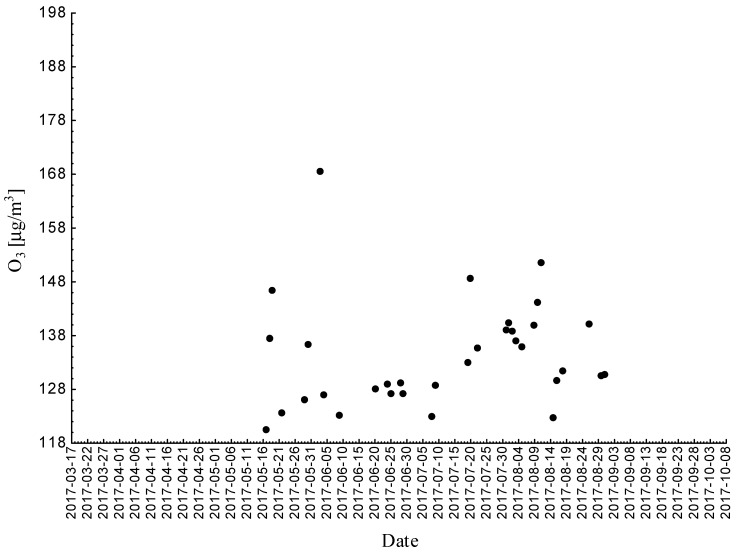
Maximum daily 8-h mean ozone concentration not less (equal to or more than) than 120.4 μg/m registered at the measurement station in Katowice (SL09KA) in 2017.

**Figure 8 ijerph-18-01473-f008:**
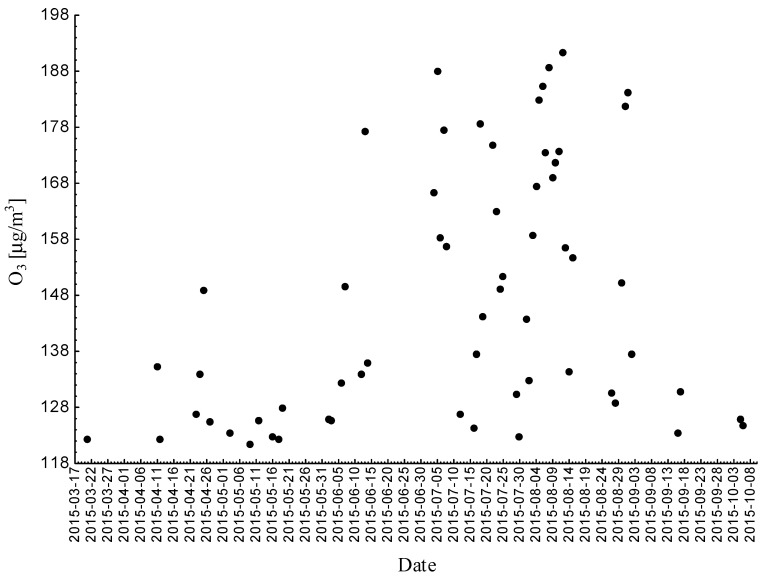
Maximum daily 8-h mean ozone concentration not less (equal to or more than) than 120.4 μg/m^3^ registered at the measurement station in Zloty Potok (SL03ZP) in 2015.

**Figure 9 ijerph-18-01473-f009:**
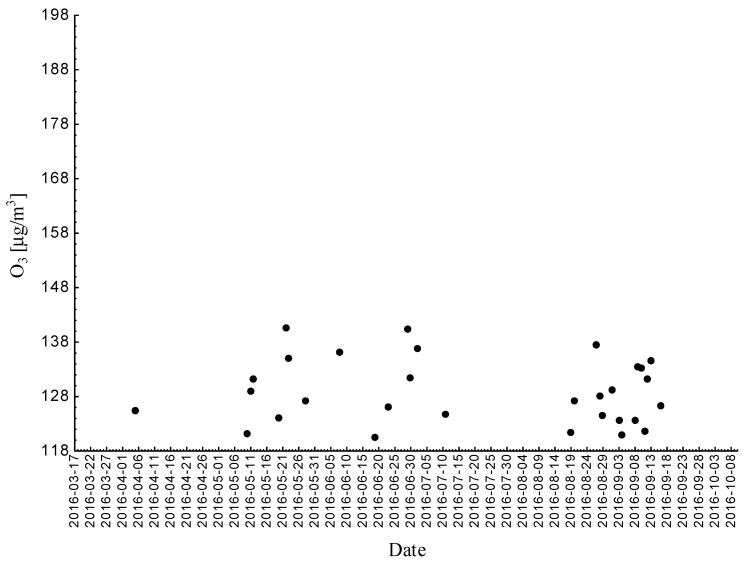
Maximum daily 8-h mean ozone concentration not less (equal to or more than) than 120.4 μg/m^3^ registered at the measurement station in Zloty Potok SL03ZP in 2016.

**Figure 10 ijerph-18-01473-f010:**
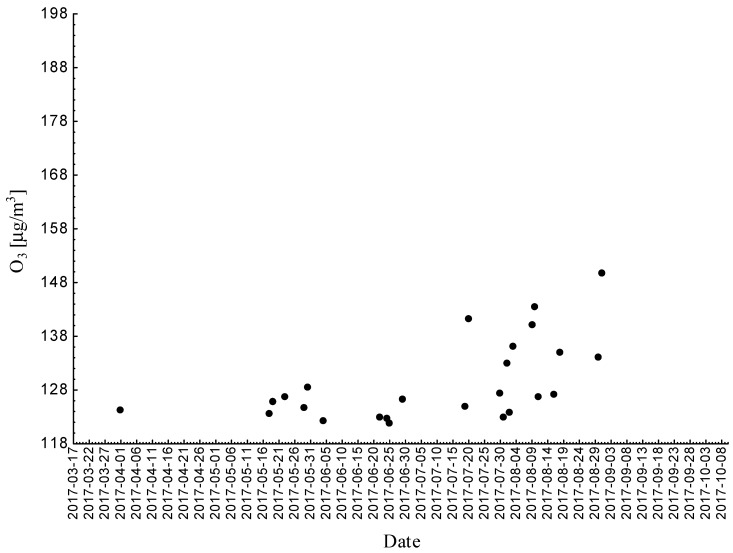
Maximum daily 8-h mean ozone concentration not less (equal to or more than) than 120.4 μg/m registered at the measurement station in Zloty Potok SL03ZP in 2017.

**Table 1 ijerph-18-01473-t001:** WHO air quality guidelines for ozone (8-h concentrations) based on time-series research [38].

	Daily Maximum 8-h Mean	Health Effects for Selected Level Exposure
High/alarm level	240 µg/m^3^	Acute health effects mainly affect vulnerable populations
Interim target	160 µg/m^3^	Significant health effects; does not provide sufficient protection for the general population Exposure to this level of ozone is related to: Inflammatory and physiological lung effects in healthy exercising young adults exposed for 6.6 h Health effects in children An estimated 3–5% increase in daily mortality
Air quality recommendation	100 µg/m^3^	Ensures sufficient protection for the general population although some health effects may occur below this level. Exposure to this level of ozone is related to: An assessed 1–2% rise in daily mortality Probability that real exposure can be repeated Health effects in children and highly sensitive and clinically endangered persons Probability that ambient ozone is a criterion for related oxidants

**Table 2 ijerph-18-01473-t002:** Ozone concentration standards (in μg/m^3^) based on national and EU legislation, WHO and EPA recommendation.

	Time	WHO	EPA	EU	Poland
Ozone concentration admissible limits	Maximum daily 8-h mean	100 µg/m^3^	0.070 ppm *	120 µg/m^3^, 25 days averaged over 3 years	120 µg/m^3^, not to be exceeded more than 25 times a calendar year

* The EPA standard for ozone is based on ppm (1 ppm = 1000 mg/m^3^ 100 µg = 0.10000 mg).

**Table 3 ijerph-18-01473-t003:** Exceedances of the target level of ozone (maximum daily 8-h mean equal to or more than 120 μg/m^3^) in period 2015–2017—number of days.

Years	2015	2016	2017
Measuring stations and number of days with exceedances			
SL15BB	32	8	15
SL09KA	32	15	32
SL03ZP	61	30	25

**Table 4 ijerph-18-01473-t004:** (**a**) The Chebyshev distances between the features O_3_ (μg/m^3^), max 8 h, for period 2015–2017. (**b**) The Chebyshev distances between the features O_3_ (μg/m^3^), max 8 h, for years 2015, 2016 and 2017.

(**a**)			
**Variable**	**Chebyshev Distances in Period 2015–2017**		
	**SL15BB (O_3_) (μg/m^3^)-Max 8 h**	**SL09KA (O_3_) (μg/m^3^)-Max 8 h**	**SL03ZP (O_3_) (μg/m^3^)-Max 8 h**
SL15BB (O_3_) (μg/m^3^)-Max 8 h	0.0	55.4	73.8
SL09KA (O_3_) (μg/m^3^)-Max 8 h	55.4	0.0	78.4
SL03ZP (O_3_) (μg/m^3^)-Max 8 h	73.8	78.4	0.0
(**b**)			
**Variable**	**Chebyshev Distances in 2015**		
	**SL09KA (O_3_) (μg/m^3^)-Max 8 h**	**SL09KA (O_3_) (μg/m^3^)-Max 8 h**	**SL03ZP (O_3_) (μg/m^3^)-Max 8 h**
SL15BB (O_3_) (μg/m^3^)-Max 8 h	0.0	47.9	73.8
SL09KA (O_3_) (μg/m^3^)-Max 8 h	47.9	0.0	78.4
SL03ZP (O_3_) (μg/m^3^)-Max 8 h	73.8	78.4	0.0
**Variable**	**Chebyshev distances in 2016**		
	**SL15BB (O_3_) (μg/m^3^)-Max 8 h**	**SL09KA (O_3_) (μg/m^3^)-Max 8 h**	**SL03ZP (O_3_) (μg/m^3^)-Max 8 h**
SL15BB (O_3_) (μg/m^3^)-Max 8 h	0.0	49.8	48.2
SL09KA (O_3_) (μg/m^3^)-Max 8 h	49.8	0.0	41.0
SL03ZP (O_3_) (μg/m^3^)-Max 8 h	48.2	41.0	0.0
**Variable**	**Chebyshev distances in 2017**		
	**SL15BB (O_3_) (μg/m^3^)-Max 8 h**	**SL09KA (O_3_) (μg/m^3^)-Max 8 h**	**SL03ZP (O_3_) (μg/m^3^)-Max 8 h**
SL15BB (O_3_) (μg/m^3^)-Max 8 h	0.0	55.4	70.6
SL09KA (O_3_) (μg/m^3^)-Max 8 h	55.4	0.0	76.2
SL03ZP (O_3_) (μg/m^3^)-Max 8 h	70.6	76.2	0.0

**Table 5 ijerph-18-01473-t005:** (**a**) Cluster analysis for period 2015–2017. (**b**) Cluster analysis for year 2015. (**c**) Cluster analysis for year 2016. (**d**) Cluster analysis for year 2017.

(**a**)		
**Cluster Number**	**Distance Euclidean Clusters—2015–2017 Distance under Diagonal Square Distance above Diagonal**	
	**No 1**	**No 2**
No 1	0.0	313.3
No 2	17.7	0.0
Variable—2015–2017	Elements cluster number 1 and distance from center right cluster. Cluster has 1 variable	
	Distance	
SL03ZP (O_3_) (μg/m^3^)-Max 8 h	0.0	
Variable—2015–2017	Elements cluster number 2 and distance from center right cluster. Cluster has 2 variables	
	Distance	
SL15BB (O_3_) (μg/m^3^)-Max 8 h	7.1	
SL09KA (O_3_) (μg/m^3^)-Max 8 h	7.1	
(**b**)		
**Cluster Number**	**Distance Euclidean Clusters—2015 Distance under Diagonal Square Distance above Diagonal**	
	**No 1**	**No 2**
No 1	0.0	491.6
No 2	22.2	0.0
Variable—2015	Elements cluster number 1 and distance from center right cluster. Cluster has 1 variable	
	Distance	
SL03ZP (O_3_) (μg/m^3^)-Max 8 h	0.0	
Variable-2015	Elements cluster number 2 and distance from center right cluster. Cluster has 2 variables	
	Distance	
SL15BB (O_3_) (μg/m^3^)-Max 8 h	7.6	
SL09KA (O_3_) (μg/m^3^)-Max 8 h	7.6	
(**c**)		
**Cluster Number**	**Distance Euclidean Clusters—2016 Distance under Diagonal Square Distance above Diagonal**	
	**No 1**	**No 2**
No 1	0.0	206.5
No 2	14.4	0.0
Variable—2016	Elements cluster number 1 and distance from center right cluster Cluster has 1 variable	
	Distance	
SL03ZP (O_3_) (μg/m^3^)-Max 8 h	0.0	
Variable—2016	Elements cluster number 2 and distance from center right cluster. Cluster has 2 variables	
	Distance	
SL15BB (O_3_) (μg/m^3^)-Max 8 h	6.5	
SL09KA (O_3_) (μg/m^3^)-Max 8 h	6.5	
(**d**)		
**Cluster Number**	**Distance Euclidean Clusters-2017 Distance under Diagonal Square Distance above Diagonal**	
	**No 1**	**No 2**
No 1	0.0	248.3
No 2	15.8	0.0
Variable—2017	Elements cluster number 1 and distance from center right cluster. Cluster has 1 variable	
	Distance	
SL03ZP (O_3_) (μg/m^3^)-Max 8 h	0.0	
Variable—2017	Elements cluster number 2 and distance from center right cluster. Cluster has 2 variables	
	Distance	
SL15BB (O_3_) (μg/m^3^)-Max 8 h	7.2	
SL09KA (O_3_) (μg/m^3^)-Max 8 h	7.2	

**Table 6 ijerph-18-01473-t006:** Emission of ozone precursors in Silesian Voivodship in 2015 based on the Silesian Regional Assembly 2016 [47].

Type of Emission	Emissions of Ozone-Forming Gases (Ground-Level Ozone Precursors) in Tons			
	CO	SO₂	Non-NH_3_ Volatile Organic Compounds	NH_3_
Burning coal for heating of households (residential plants)	255,499.186	26,308.687	26,448.750	153.928
Transportation	18,579.824	156.445	2236.275	
Point sources	16,351.951	64,336.607	2516.725	441.240
Agricultural activities	988.525	2.151	3990.702	8389.749
Natural sources (forests)			13,282.668	1504.178
Total	439,419.486	90,803.890	48,475.120	10,489.095

**Table 7 ijerph-18-01473-t007:** Emission of NOx in 2015 [47].

Type of Emission	NOx
Burning coal for heating of households	9145.177
Transportation	7296.671
Point sources	46,893.604
Agricultural activities	1855.560
Total	65,191.012

**Table 8 ijerph-18-01473-t008:** Selected Nomenclature for sources of Air Pollution (SNAP) 97 classification groups [49].

SNAP97 Groups	Sources of Emission
SNAP 01	Combustion in energy and transformation industries
SNAP 02	Non-industrial combustion plants
SNAP 0202 subgroup	Residential plants
SNAP 03	Combustion in manufacturing energy
SNAP 04	Production processes
SNAP 05	Extraction and distribution of fossil fuels and geothermal energy
SNAP 06	Solvent and other product use
SNAP 07	Road transport
SNAP 08	Other mobile sources and machinery
SNAP 09	Waste treatment and disposal
SNAP 10	Agriculture
SNAP 11	Other sources and sinks

**Table 9 ijerph-18-01473-t009:** Emission of ozone precursors in Silesian voivodships in 2018 based on [48].

Source of Emission	SNAP97 Groups	Emissions of Ozone-Forming Gases (Ground-Level Ozone Precursors) in tones			
		CO	SO₂	Non-NH_3_ Volatile Organic Compounds	NH_3_
Residential plants	0202	264,356.59	22,469.00	28,118.42	
Manufacturing and energetics	01	10 934.01	33,835.37	90.99	32.24
	02	2005.06	627.00	224.79	
	03	3149.24	2644.53	41.26	250.01
	04	148,768.17	8954.02	1817.67	28.25
	05	38.75	76.86	155.08	
	06	284.34	3.05	1916.11	2.68
	09	86.03	5.54	12.22	7.86
Road transport	07	54,724.62	52.69	7753.86	446.85
Agricultural tractors and other agricultural machinery	08	3156.58	6.86	243.06	0.55
Rail transport	08	69.75	0.65	30.31	0.05
Airports	08	38.60	10.15	5.17	
Spoil tips and excavatability	05				
Waste storage	09				
Agriculture	10			2673.93	6305.08
Forestry and grounds	11				
Total		487,611.74	68,685.72	43,127.87	7073.57

**Table 10 ijerph-18-01473-t010:** Emission of NOx in 2018 [48].

Type of Emission	SNAP97 Groups	NOx
Residential plants	0202	7167.04
Manufacturing and energetics	01	30,325.00
	02	546.42
	03	4732.41
	04	9380.07
	05	52.25
	06	93.10
	09	40.73
Road transport	07	28,671.47
Agricultural tractors and other agricultural machinery	08	2364.48
Rail transport	08	341.59
Airports	08	127.75
Spoil tips and excavatability	05	
Waste storage	09	
Agriculture	10	
Forestry and grounds	11	
**Total**		83,842.31

## Data Availability

Not applicable.

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
