# Peer review of "Ozone Concentration Levels in Urban Environments—Upper Silesia Region Case Study"

_ijerph, 2021, doi:10.3390/ijerph18041473_

Round 1

Reviewer 1 Report

The article is ready to be published 

Reviewer 2 Report

After going through the revised text I am of the impression that the paper is now significantly improved and can be published in its current form.

Reviewer 3 Report

Authors have addressed the comments effectively.

This manuscript is a resubmission of an earlier submission. The following is a list of the peer review reports and author responses from that submission.

Round 1

Reviewer 1 Report

The scientific theme addressed in this article is a current and relevant academic theme, but it does not satisfactorily explore some of the most important bibliographies in the area (the classics), although it also presents ancient literature (today we have more recent works). The structure of the proposed article is adequate and of good quality, associating the results with the text, but not as a research proposal (it simply made a tabulation of data). This point is clear, especially when considering the literature, although the methodology presented does not present any novelty, it does not present any relevant statistical treatment and also when it describes health risks, there are bibliographic revisions, although in the discussions he comments on overcoming the levels of ozone concentrations within the limits stipulated by WHO and EPA. For the region, the article may become important in modeling pollution data (the author has not explored that the region is an important indicator of ozone formation and who produces it). The author raises a series of data and makes a superficial analysis of whether the limit of ozone concentrations has been reached or not. Therefore, in my personal point of view, the authors demonstrated a document that is not very consistent where they obtain the results, but do not explore scientifically, showing new methodologies. The discussion and conclusion are conveniently linked to the object of the research, but it does not add scientific values ​​to the proposal of the article, which was proposed as a research question, in order to apply the theory to a consistent work of scientific research. Thus, all these considerations, allow me not to recommend approval for the direct publication of this document, correspond to the requirements of the scientific works that will serve as an indicator for public policy decision making.

regarding the detection of plagiarism I suggest the editors submit the article to software with these purposes

Reviewer 2 Report

I consider this paper to be of great interest to the readers. I have the following comments before I recommend publication of the paper:

  1. The motivation behind the study may be stated in a more extensive manner;
  2. The newness of the study should be clearly stated in the conclusion;
  3. The following relevant references can improve the literature survey:

Spectral analysis approach to study the association between total ozone concentration and surface temperature, (2020) International Journal of Environmental Science and Technology, 17 (10), pp. 4353-4358.

Information Theoretic Study of the Ground-Level Ozone and Its Precursors Over Kolkata, India, During the Summer Monsoon (2020) Iranian Journal of Science and Technology, Transaction A: Science, DOI: 10.1007/s40995-020-01007-x

Reviewer 3 Report

The topic is certainly worthy of analysis, given the implications for the environment and human health.

I believe that the research approach should be improved by including different methods of data analysis. For example, those based on machine learning, such as in Riyang Liu et al., Environment International, 142, 2020; Xiaoqian Su et al., Atmospheric Pollution Research, 11, 2020; Vairo T. et al., Chemical Engineering Transactions, 82, 2020 and Chemical Engineering Transactions, 74, 2019.

Consequently, I suggest to include a sensitivity analysis on the factors that affect the concentration of ozone, in order to depict a simplified assessment on air-quality decision-making process, by analyzing the different plausible scenarios.
